

# Age, growth, mortality and reproductive seasonality of jolthead porgy, *Calamus bajonado*, from Florida waters

Michael L. Burton[*], Jennifer C. Potts[*], Jordan Page and Ariel Poholek

National Marine Fisheries Service, Beaufort Laboratory, Beaufort, NC, USA
[*] These authors contributed equally to this work.

## ABSTRACT

Ages of jolthead porgy (*Calamus bajonado* Schneider 1801) ($n = 635$) from Florida commercial and recreational fisheries from 2008–2016 were determined using sectioned sagittal otoliths. We determined, using edge-type analysis, that opaque zones were annular, forming March–June (peaking in April). Jolthead porgy ranged from 1–13 years, and the largest fish measured 680 mm TL (total length, mm). Body size relationships for jolthead porgy were $TL = 1.09FL + 20.44$ ($n = 622$, $r^2 = 0.99$), $FL = 0.90\,TL - 14.26$ ($n = 622$, $r^2 = 0.99$), and $W = 1.1 \times 10^{-5}\,TL^{3.06}$ ($n = 577$), where $W$ is total weight (grams, g) and FL is fork length (mm). The von Bertalanffy growth equation for jolthead porgy was $L_t = 737(1 - e^{-0.14(t+2.02)})$ ($n = 635$). Point estimate of natural mortality was $M = 0.32$, while age-specific estimates of $M$ ranged from 0.58–0.17 $y^{-1}$ for ages 1–13. Catch curve analysis estimated the instantaneous rate of total mortality $Z = 0.70$, while instantaneous rate of fishing mortality $F$ was 0.38. Macroscopic staging of female gonads indicated the presence of hydrated oocytes from December–March, and GSI data indicates that peak spawning in females occurs during March. This study presents the first published findings of life history parameters for jolthead porgy from the Atlantic waters off the southeastern United States.

Corresponding author
Michael L. Burton,
michael.burton@noaa.gov

## INTRODUCTION

Jolthead porgy (*Calamus bajonado* Schneider 1801) is one of the largest porgies (Family Sparidae) in the western Atlantic, capable of attaining lengths of almost 800 mm total length (TL). The species is distributed widely in the western Atlantic Ocean, from Rhode Island southward to Brazil, including Bermuda and throughout the Caribbean and into the Gulf of Mexico to northwest Florida (*Randall & Caldwell, 1966*), although it typically is not found in fishery catches north of Cape Canaveral, Florida (*Dixon & Huntsman, 2004*). Adults normally inhabit subtropical to tropical waters of the coastal and insular shelfs at depths from 0–200 m (*Smith, 1997*). Juveniles or sub-adults have been collected from seagrass beds in the U. S. Virgin Islands (*Randall & Caldwell, 1966*). Like many members of the family Sparidae, the species exhibits protogynous hermaphrodism (*Dubovitskij, 1974*, in *Claro, Lindeman & Parenti, 2001*).

Jolthead porgy is not a target species of reef fisheries in the southeastern United States (SEUS, North Carolina to Florida Keys, including the Dry Tortugas) but is an important secondary species in the snapper-grouper fishery. The following raw landings data may be found in the supplemental data file, in the Landings tab. Estimated annual landings from headboats (vessels carrying at least seven anglers engaged in recreational fishing) sampled by the Southeast Region Headboat Survey (SRHS) averaged 1,886 fish annually from 1981–2015, with an average of 62% of those coming from Florida waters. Estimated annual numbers of fish landed in the SEUS by anglers fishing from private recreational boats and charter boats averaged 73,381 fish from 1981–2015, with 98% of those landed in Florida waters. Commercial fisheries landings from the SEUS are exclusively from Florida, averaging 435 kg annually from 1996–2014. We posit that the lack of commercial landings from outside Florida may be due to non-retention of this less important species by commercial fishermen, while lack of private recreationally caught jolthead porgy might be attributable to a species identification issue (lumping of jolthead porgy with other porgy species). However, we queried the Southeast Reef Fish Survey (SERFS) database, a long-term chevron trap fishery-independent survey that samples the SEUS, for occurrences of jolthead porgy being captured in traps deployed off North Carolina or South Carolina, and found no instances of capture in the traps back to 1990. Fisheries landings are temporally variable with no apparent increasing or decreasing trends.

Jolthead porgy is currently included in the South Atlantic Fishery Management Council's Snapper-Grouper Fishery Management Plan (*SAFMC, 2017*). Jolthead porgy is managed by inclusion in the 20-fish aggregate bag limit for recreational fisheries. Recreational catches are further regulated by a porgy complex (jolthead porgy; knobbed porgy *Calamus nodosus* Randall & Caldwell 1966; Whitebone porgy *Calamus leucosteus* Jordan and Gilbert 1885; saucereye porgy *Calamus calamus* Valenciennes 1830; and scup *Stenotomus chrysops* Linnaeus 1766) aggregate annual catch limit (ACL), or quota, of 48,495 kg (*SAFMC, 2017*). The commercial fishery is managed with an ACL of 16,487 kg for the porgy complex.

The recreational quota for the porgy complex was reached on September 17, 2014, triggering the first closure of the recreational fishery for those five species. While there was limited life history information on some of these species in SEUS waters (knobbed porgy—*Borden, 2001*; *Horvath, Grimes & Huntsman, 1990*; whitebone porgy—*Waltz, Roumillat & Wenner, 1982*; *Sedberry, 1989*) there were no published studies on jolthead porgy to assist fishery managers in determining appropriate ACLs. This lack of basic biological information on an important species was a prime motivator behind this study. Knowledge about growth characteristics and natural mortality are important inputs into stock assessments that lead to the development of annual quotas, or overfishing limits (*Carruthers et al., 2017*). We used sagittal otoliths and gonad samples collected through dockside sampling programs to provide the first published information on age-growth parameters, natural mortality rates and reproductive seasonality for jolthead porgy in the SEUS, filling in an important gap for this data-poor species.

## MATERIALS AND METHODS

### Sampling

Jolthead porgy ($n = 630$) were opportunistically sampled from fisheries landings by NMFS and state agencies' port agents sampling the recreational and commercial fisheries along the SEUS coast during 2015 and 2016. Additionally we used eight (8) archived otolith samples collected by the SRHS in 2008 and 2011. Sampling occurred year round, All samples came from Florida, with southeast Florida (Ft. Pierce through Miami) accounting for 30% ($n = 190$) and the Florida Keys accounting for 70% ($n = 448$). The majority of samples were collected from fish caught by headboat anglers ($n = 546$), with private recreational anglers/charterboat anglers ($n = 7$) and the commercial fisheries sector ($n = 85$) accounting for the rest. All specimens were captured by either conventional vertical hook and line gear or divers with spears. Total lengths (TL) and/or fork lengths (FL) of specimens were recorded in millimeters (mm). Whole weight (W) in grams was recorded for fish landed in the recreational headboat fishery. Fish landed by commercial fisheries were eviscerated at sea and were excluded from the W–TL regression analysis. Sagittal otoliths were removed during dockside sampling and stored dry in coin envelopes.

### Reproductive seasonality

Gonads were collected during 2015–2016 from fish sampled from headboats, as commercially caught fish were eviscerated at sea prior to sampling. Sex was assigned using macroscopic examination of the gonads (rounded cross section, females; triangular cross section, males) and recorded. Whole gonads were weighed to the nearest 0.01 grams, macroscopically staged using the general terminology of *Brown-Peterson et al. (2011)* and preserved in 10% buffered formalin for future histological processing and determination of timing of transition from female to male, which cannot be done reliably macroscopically. Reproductive seasonality was assessed for this study using a gonadosomatic index (GSI) for females, calculated as

$$GSI = (gonad\ weight/whole\ weight) * 100.$$

### Age determination and timing of opaque zone formation

Otoliths were sectioned on a low-speed saw, following the methods of *Potts & Manooch III (1995)*. Three serial 0.5 mm sections were taken from the transverse plane near the otolith core. The sections were mounted on microscope slides with thermal cement and covered with mounting medium before analysis. The sections were viewed under a dissecting microscope at 12.5X using transmitted light.

Age determination was based on recording an opaque zone count and edge type code by an experienced reader (MLB) with extensive experience interpreting otolith sections (*Burton, 2001*; *Potts, Burton & Myers, 2016*) with no knowledge of date of capture or fish size for each sample. The edge codes refer to the type of zone, opaque or translucent, and in the case of translucent zones the amount of that zone between the last opaque zone formed and the otolith section edge. The codes used are outlined below:

1 = opaque zone forming on the edge of the otolith section;

2 = narrow translucent zone on the edge, generally <30% of the width of the previous translucent zone;

3 = moderate translucent zone on the edge, generally 30%–60% of the width of the previous translucent zone;

4 = wide translucent zone on the edge, generally >60% of the width of the previous translucent zone (*Harris et al., 2007*).

To ensure consistency in interpretation of the growth zones on the otolith, a subset of the otolith sections ($n = 520$; 82%) were then read by a second reader (JP), and an index of between-reader average percent error (APE) was calculated, following the methodology of *Beamish & Fournier (1981)*. Where readings for a specimen disagreed, the sections were viewed again together. If consensus was reached the sample was retained; otherwise, it was excluded from further analysis.

Timing of opaque zone formation was assessed using edge analysis. The edge types were plotted by month of capture to determine if the opaque zones were deposited primarily in one season or month. Based upon edge frequency analysis, all samples were assigned a calendar age, obtained by increasing the opaque zone count by one if the fish was caught before that year's opaque zone was formed and had an edge which was a moderate to wide translucent zone (type 3 or 4). Fish caught during the time of year of opaque zone formation with an edge type of 1 or 2, as well as fish caught after the time of opaque zone formation, were assigned a calendar age equivalent to the opaque zone count.

## Growth

Growth of jolthead porgy was estimated using the *Von Bertalanffy (1938)* growth model:
$L_t = L_{inf}(1 - e^{(-K(t-t_0))})$ where $L_{inf}$ = theoretical maximum length, $K$ = Brody growth coefficient (rate at which maximum size is attained), and $t_0$ = theoretical age at size 0. Parameters were estimated from observed length at age using AD Model Builder estimation software (Otter Research Ltd., Sidney, B.C., Canada) [Mention of trade names or commercial companies is for identification purposes only and does not imply endorsement by the National Marine Fisheries Service, NOAA]. To account for growth of the fish throughout the year before or after its "birthday", the calendar age of the fish ($Age_c$) was adjusted for the time of year caught ($Mo_c$) compared to month of peak spawning, or "birthdate", ($Mo_b$), determined from the reproductive component of this study, thus creating a fractional, or monthly biological age, ($Age_f$):

$$Age_f = Age_c + ((Mo_c - Mo_b)/12).$$

## Body-size relationships

For weight–length relationships we regressed $W$ on TL and FL ($n = 577$) using all fish with both lengths and whole weights sampled for this study, examining both a non-linear fit using nonlinear least squares estimation (version 9.4; SAS Institute, Cary, NC, USA) and a linearized fit of the log-transformed data. Residuals were examined to determine which regression provided the best fit. For length-length relationships, we regressed TL on FL and FL on TL ($n = 622$) using linear regression.

## Natural mortality

We estimated the instantaneous rate of natural mortality ($M$) of jolthead porgy using several methods:

(1) *Hewitt & Hoenig*'s *(2005)* longevity mortality relationship

$$M = 4.22/t_{\mathrm{max}}$$

where $t_{\mathrm{max}}$ is the maximum age of the fish in the sample;

(2) *Charnov, Gislason & Pope*'s *(2013)* method using life history parameters

$$M = (L/L_\infty)^{-1.5}K$$

where $L_\infty$ and $K$ are the von Bertalanffy growth equation parameters and $L$ is fish length at age;

(3) *Then et al.*'s *(2015)* $t_{max}$-based estimator

$$M = 4.899 t_{max}^{-0.916};$$

and (4) *Then et al.*'s *(2015)* growth-based estimator

$$M = 1.118 K^{0.73} L_\infty^{-0.33}.$$

The equation of *Hewitt & Hoenig (2005)* and the two equations from *Then et al. (2015)* use either maximum age or von Bertalanffy growth parameters to generate a single point estimate. The *Charnov, Gislason & Pope (2013)* method, which incorporates life history information via the growth parameters, is based upon evidence suggesting that $M$ decreases as a power function of body size. This method generates age-specific rates of $M$ and has recently been used in Southeast Data Assessment and Review (SEDAR) stock assessments (*SEDAR, 2017a*).

The estimated percent of the population to survive to the oldest observed age in the population was calculated with the natural mortality rates applied to the fully-recruited ages in the fishery (modal age + 1). The following equation was used:

$$\%\mathrm{survivorship} = 100 * (\exp(-\sum (M\mathrm{age_f} - M\mathrm{age_o}))),$$

where $M\mathrm{age_f}$ = natural mortality at the first age of full recruitment and $M\mathrm{age_o}$ = natural mortality at the oldest age in the population.

## Total and fishing mortality

The total instantaneous mortality rate $Z$ was estimated using a catch curve analysis of the age-frequency of the samples (*Beverton & Holt, 1957*). We used only fully recruited ages (modal age plus one), since the age group at the top of the catch curve may not yet be fully vulnerable to the fishing gear (*Everhart, Eipper & Youngs, 1975*). The instantaneous rate of fishing mortality, $F$, was estimated by subtracting $M$ from $Z$.

# RESULTS

## Reproductive seasonality

Gonads were collected from 178 individuals (22 males, 156 females). Due to lack of samples we did not analyze male reproductive seasonality. Jolthead porgy exhibit increasing ovary

development from December to March, with peak spawning in March, as indicated by gonadosomatic index (Fig. 1A). This trend is supported by a plot of relative proportion of gonad stage by month (Fig. 1B), showing fish with primarily hydrated oocytes from December–March, followed in subsequent months by a preponderance of fish in a regressing or resting gonad stage. The lack of gonad samples from June through September is likely due to a combination of two factors. First, the species was not encountered as frequently by samplers during the months of June through August. The reason for this temporal absence in the landings is unknown, but it is apparent in the otolith collections as well (Fig. 2). Secondly, it is probable that any jolthead porgy that were encountered were in a resting/regressing stage and the gonads were overlooked. This is a plausible explanation for why no gonad samples were taken in September, when sample sizes rebounded.

## Age determination and timing of opaque zone formation

We sectioned 638 jolthead porgy sagittae collected. Opaque zones were counted on 635 (99.5%) jolthead porgy sections. Three samples were determined to be illegible and excluded from the analyses.

We were able to assign an edge type to 99% of our samples ($n = 633$) for our analysis of opaque zone formation timing. Jolthead porgy otoliths exhibited opaque zones on the margin March–June, with a peak in April (Fig. 2). Opaque zones on the edge were absent from July through February. A shift to narrow translucent edge was noted from May to September. Moderate to wide translucent edges were found October through January, and the widest translucent edge was found during February and March, prior to peak opaque zone formation in April. We assumed that opaque zones on jolthead porgy otoliths were annuli.

Calendar ages were assigned as follows: for fish caught January through June and having an edge type of 3 or 4, the annuli count was increased by one; for fish caught in that same time period with an edge type of 1 or 2, calendar age was equivalent to annuli count; for fish caught from July to December, the calendar age was equivalent also to the annuli count.

Opaque zones on jolthead porgy otoliths were clear and easy to interpret (Fig. 3) and edge types were assigned consistently between readers. Between-reader APE was 0.06% ($n = 520$), meeting *Campana*'s *(2001)* acceptable value for APE (5% for species of moderate longevity and reading complexity). In fact, the readings only disagreed on two of 520 samples and only by ± one year in both cases. Both readers noted that those two otolith samples were of marginal quality. Consensus on between-reader counts was reached on all samples, and no samples were excluded from further analyses. The opaque zones were easy to trace from the sulcal groove out into the lateral plane on the dorsal side of the sections. The opaque zones were also matched with those on the ventral side, but due to refraction of the crystalline matrix on the ventral portion, the most consistent counts were made on the dorsal portion.

## Growth

Growth of jolthead porgy was modeled using fractional ages based on the birth month, or month of peak spawning, of March determined from the reproductive samples collected.

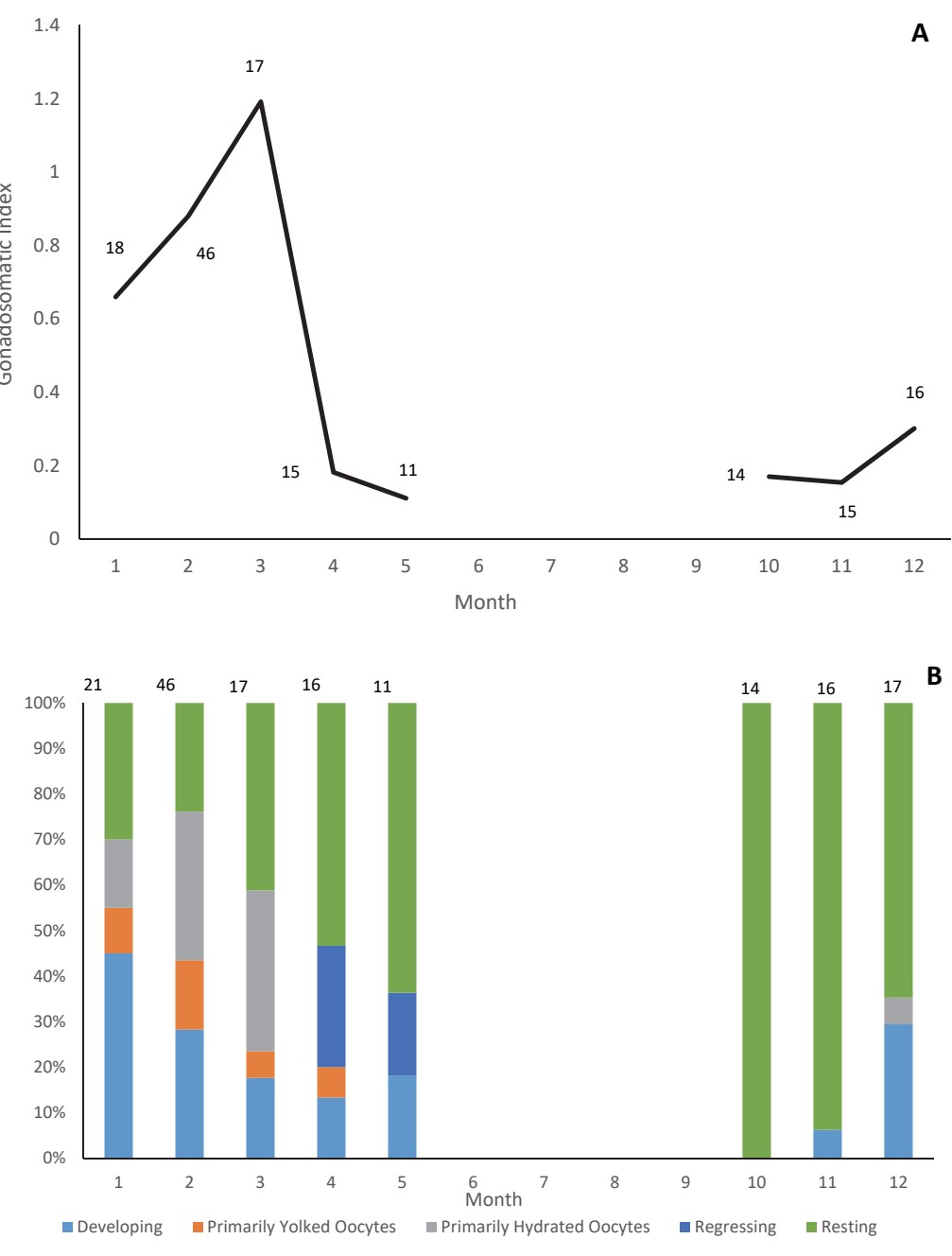

**Figure 1  Reproductive seasonality of jolthead porgy.** Examination of reproductive seasonality of jolthead porgy (*Calamus bajonado*) collected from Florida in 2015–2016 as determined by (A) mean monthly gonadosomatic index of females sampled, and (B) relative proportion of female gonadal development by month, determined macroscopically. Sample size for each month is noted on the figure with the associated graphical point.

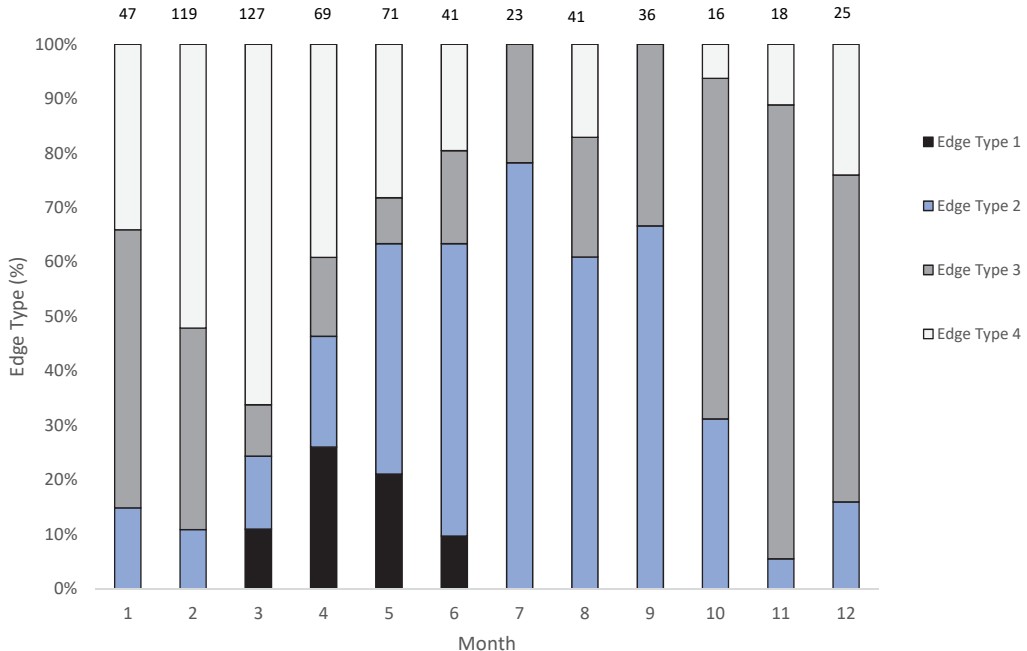

**Figure 2  Analysis of edge type of otoliths of jolthead porgy.** Monthly percentages of all edge types for jolthead porgy (*Calamus bajonado*) collected from Florida in 2008–2016. Edge codes: 1, opaque zone on edge, indicating annulus formation; 2, small translucent zone, <30% of previous translucent zone; 3, moderate translucent zone, 30–60% of previous translucent zone; 4, wide translucent zone, >60% of previous translucent zone. Sample sizes are shown above columns.

Jolthead porgy in this study ranged from 235–680 mm TL and ages 1–13, but only five fish were older than age-8 (Table 1). The standard deviation of length-at-age increased with age, but the coefficient of variation (CV) remained relatively constant. Thus, we assumed constant CV in our model estimation. Growth is described by the following equation:

$$L_t = 737(1 - e^{-0.14(t+2.02)})\,(n = 635;\ \text{Fig. 4}).$$

While our data included large numbers of aged 2–4 fish, we had just one fish smaller than 250 mm TL. This lack of smaller fish is likely explained by gear selectivity, as our samples were all fishery-dependent and fish smaller than this size are likely unable to recruit to the hooks used in the fishery, or by fishers not retaining small fish in their catch, especially in light of the bag-limit regulations. Therefore we re-estimated the growth model using the method of *McGarvey & Fowler (2002)*, which uses a left-truncated normal probability density function of length to adjust for the bias imposed by minimum size limits (or some other selectivity imposed by the fisheries) by assuming a zero probability of capture below the minimum size limit. We assumed a de facto minimum size limit for this analysis of 250 mm TL and estimated the growth model parameters with $t_0$ freely estimated by the model. This procedure has the effect of pulling the growth curve down to simulate smaller fish length-at-age for the youngest ages. We applied the full, untruncated normal likelihood to specimens in the study not subject to the minimum size limit. Parameters were estimated by minimizing the negative sum of log-likelihoods using AD Model Builder estimation

A.

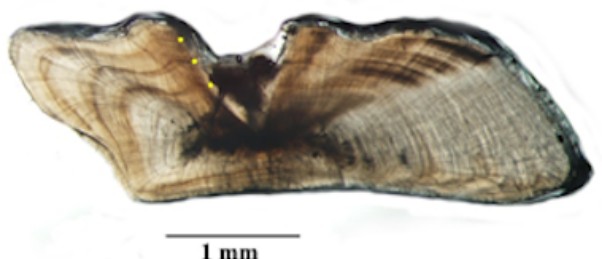

1 mm

B.

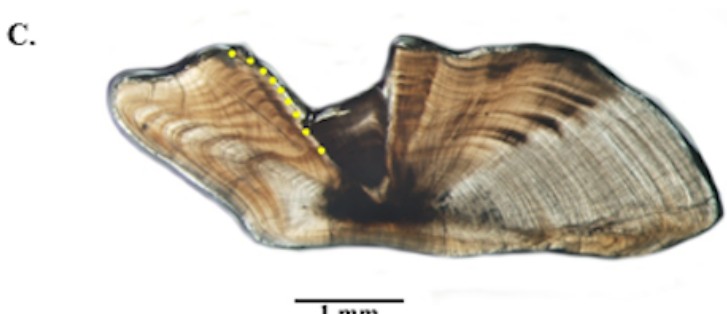

1 mm

C.

1 mm

**Figure 3** **Photographs of jolthead porgy otolith sections.** Sections from otoliths of jolthead porgy (*Calamus bajonado*): (A) 397 mm TL 3-yr old; (B) 527 mm TL 6-yr old; (C) 570 mm TL 9-yr old. Age was determined by counting opaque zones along the dorsal axis and sulcus using transmitted light at 12.5× magnification.

**Table 1** Observed and predicted (*Von Bertalanffy, 1938*) mean total length (TL), measured in millimeters, and natural mortality at age (*M*; *Charnov, Gislason & Pope, 2013*) data for jolthead porgy (*Calamus bajonado*) collected from 2008–2016 along the Florida coast. Standard errors of the means (SE) are provided in parentheses.

| Age | *n* | Mean TL (±SE) | TL range | Predicted TL | *M* y$^{-1}$ |
|-----|-----|---------------|----------|--------------|--------------|
| 1 | 18 | 300 (8) | 250–375 | 254 | 0.58 |
| 2 | 235 | 320 (2) | 235–405 | 317 | 0.44 |
| 3 | 159 | 370 (3) | 284–520 | 372 | 0.35 |
| 4 | 120 | 418 (4) | 280–535 | 420 | 0.30 |
| 5 | 46 | 452 (8) | 329–600 | 461 | 0.27 |
| 6 | 26 | 492 (11) | 380–610 | 497 | 0.24 |
| 7 | 18 | 517 (20) | 332–640 | 529 | 0.22 |
| 8 | 8 | 545 (23) | 470–632 | 556 | 0.21 |
| 9 | 2 | 523 (48) | 475–570 | 579 | 0.20 |
| 12 | 2 | 675 (5) | 670–680 | 633 | 0.17 |
| 13 | 1 | 680 | – | 647 | 0.17 |

software. The resulting growth model is:

$$L_t = 724(1 - e^{-0.14(t+1.9)}) (n = 634, \text{Fig. 4}).$$

## Body-size relationships

Statistical analyses revealed an additive error term (variance not increasing with size) in the residuals of the $W - TL$ and $W - FL$ relationships for jolthead porgy, indicating that direct non-linear fits of the data were appropriate. The relationships are described by the following equations:

$$W = 1.1 \times 10^{-5} TL^{3.05} (n = 577) \text{ and } W = 4.2 \times 10^{-5} FL^{2.90} (n = 577).$$

The relationships between TL and FL are described by the equations

$$TL = 1.09 \times FL + 20.44 (n = 622; r^2 = 0.99; p < 0.0001) \text{ and}$$

$$FL = 0.90 \times TL - 14.26 (n = 622; r^2 = 0.99; p < 0.0001).$$

## Natural mortality

Natural mortality (*M*) was estimated to be 0.32 y$^{-1}$ for jolthead porgy using *Hewitt & Hoenig*'s *(2005)* method integrating all ages into a single point estimate, using the maximum age from our study of 13 yrs. *M* was estimated to be 0.47 y$^{-1}$ using *Then et al.*'s *(2015)* $t_{max}$—based method and 0.03 y$^{-1}$ using their growth-based equation. Age-specific estimates of *M* using *Charnov, Gislason & Pope (2013)* are presented in Table 1. We used the midpoint of each age (e.g., 0.5, 1.5, 2.5, etc.) to calculate age-specific *M*, because the *Charnov, Gislason & Pope (2013)* method cannot mathematically calculate *M* for age-0. Also, for stock assessment purposes where the integer age is used to describe the entire year of the fish's life, the mid-point gives the median value of *M* for that age.

When considering the cumulative estimate of survivorship on the fully recruited age to the oldest age, the *Hewitt & Hoenig (2005)* method estimates 2.8% survivorship, while

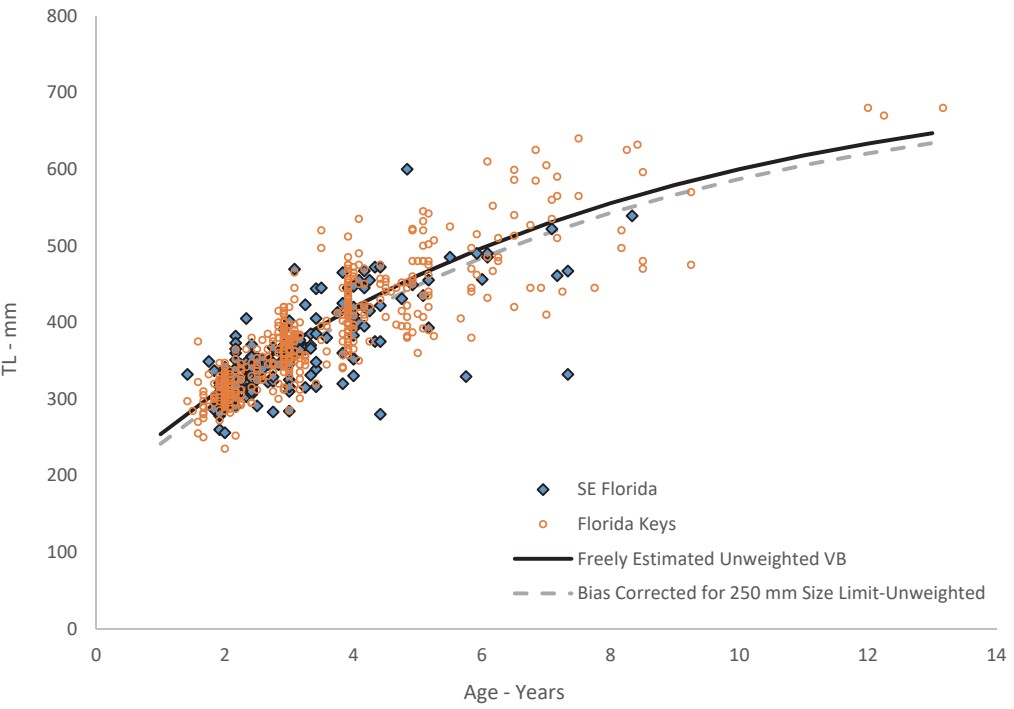

**Figure 4  Von Bertalanffy growth curves of jolthead porgy.** Von Bertalanffy growth curves of jolthead porgy (*Calamus bajonado*) collected from Florida in 2008–2016. We compare the curve corrected for bias of minimum size limits (*McGarvey & Fowler, 2002*) with the uncorrected, unweighted and freely estimated $t_0$ curve. Observed sizes at fractional ages are plotted for comparison by region of capture.

the *Charnov, Gislason & Pope (2013)* estimate is 8.3% and *Then et al.*'s *(2015)* aged-based estimate is 0.6%. When estimating survivorship only on the fully recruited ages, this estimate omits a large amount of mortality that occurs on younger fish, thus affecting the survivorship estimate. If we use all ages (0–13), survivorship is estimated to be 1.1% using *Hewitt & Hoenig (2005)* constant $M$, 1.3% using the *Charnov, Gislason & Pope (2013)* age-specific $M$, and 0.1% using *Then et al.*'s *(2015)* age based estimate.

## Total and fishing mortality

Jolthead porgy were fully recruited to the fishery by age-3. Catch curve analysis of the sample age frequency plot results in an estimate of $Z = 0.70$. If we use the *Hewitt & Hoenig (2005)* point estimate of $M$, this results in a value of fishing mortality $F = 0.38$. If we use the age-varying method of *Charnov, Gislason & Pope (2013)*, then our estimates of $F$ range from 0.35 –0.50 for ages 3–9.

## DISCUSSION

Otolith edge analysis demonstrated that jolthead porgy deposited one annulus per year from March-June, with peak annulus formation occurring in April. These results are similar to timing of annulus formation for other members of the family Sparidae in the SEUS, which tend to form annuli in the late spring-summer months (knobbed porgy –June and

July for fish from North Carolina and South Carolina, *Horvath, Grimes & Huntsman, 1990*; whitebone porgy—June and July for fish from the South Atlantic Bight, *Waltz, Roumillat & Wenner, 1982*; littlehead porgy, *Calamus proridens* Jordan and Gilbert 1884—April for fish from the eastern Gulf of Mexico, *Tyler-Jedlund & Torres, 2015*; red porgy, *Pagrus pagrus* Linnaeus 1758—March through May, with peak in April for fish from North Carolina through southeast Florida, *Potts & Manooch III, 2001*).

Growth was rapid, with jolthead porgy attaining average observed sizes of 300, 418 and 545 mm TL by ages 1, 4 and 8 respectively (Table 1). Our study contained only five fish older than age-8. Though jolthead porgy is the largest of the *Calamus* species, this fish has a similar longevity to what has been found in studies of *C. nodosus* (*Borden, 2001*), *C. leucosteus* (*Waltz, Roumillat & Wenner, 1982*) and *C. proridens* (*Tyler-Jedlund & Torres, 2015*), maximum ages of 12, 12, and 10 years, respectively, and each study also had few fish older than age-8. Two of the studies, *Waltz, Roumillat & Wenner (1982)* and *Tyler-Jedlund & Torres (2015)* collected the majority of their specimens from fishery-independent trawl surveys which fished in 11–20 m at the shallowest depths, but then supplemented their collections for larger fish from offshore hard bottom and reef habitats. These studies were able to collect small, juvenile fish (50 and 76 mm FL, respectively) from the shallow water and found that the average length of the fish increased with depth of fishing. The other studies obtained their samples from offshore commercial and recreational fishing on hard bottom or reef habitats, and were unable to collect fish smaller than ∼250 mm TL. We theorize that all four species discussed here exhibit a general ontogenetic shift from shallow water to deeper water as they grow and mature. The samples from this study were primarily from the recreational fishery. The SEUS commercial fishery tends to fish in deeper water, and if the species does move to deeper water as it gets older, as suggested by studies of other *Calamus spp.*, then the largest and oldest fish may never be available for age-growth studies, leading us to underestimate jolthead porgy's longevity or even the maximum size, 760 mm FL (849 mm TL; *Froese & Pauly, 2017*). The smallest fish in our study, 235 mm TL, was age-2 and all age-1 fish were ≥ 250 mm. This lack of small, young fish is common to studies dominated by fishery-dependent samples and can lead to problems in estimating the growth curve for the youngest ages. Because of the lack of youngest fish in this study, the corresponding growth curve for young fish should be interpreted with caution. We attempted to account for this issue by re-estimating our growth parameters using a correction to adjust for the bias imposed by fishery selectivity. This procedure had the effect of slightly reducing the initial trajectory of the growth curve (Fig. 4).

The difference in predicted sizes at the younger ages between estimation methods (Fig. 4) was slight, likely due to the number of age-1 fish in our collection. Age-0 fish were not retained in the fishery landings. The von Bertalanffy growth model does not capture growth of fish beyond the data points used to estimate the parameters. Thus, these predicted values should be used with the knowledge that they carry added uncertainty.

We compared von Bertalanffy growth parameters for the four most common *Calamus* spp. from the SEUS in Table 2. Jolthead porgy has a predicted maximum size almost twice as large as *C. nodosus*, *C. leucosteus* and *C. proridens*, though the age ranges are similar. The Brody growth coefficient *K*, the rate of attainment of maximum size, was lowest for
**Table 2  Comparison of von Bertalanffy growth parameters of *Calamus spp.* from the US Southeast region.** Standard errors of the parameter estimates for this study are provided in parentheses below the estimates.

| Species | Study | Max age | $L_\infty$ | $K$ | $t_0$ |
|---|---|---|---|---|---|
| *C. nodosus* | Borden (2001) | 12 | 357 mm FL | 0.26 | −1.81 |
| *C. leucosteus* | Waltz, Roumillat & Wenner (1982) | 12 | 362 mm FL | 0.26 | −1.40 |
| *C. proridens* | Tyler-Jedlund & Torres (2015) | 10 | 306 mm FL | 0.25 | −1.69 |
| *C. bajonado* | This study | 13 | 737 mm TL (5.67) 649 mm FL | 0.14 (0.02) | −2.02 (0.30) |

jolthead porgy, 0.14, which is about half of the values for the other species. This is expected given the inverse relationship between $L_\infty$ and $K$.

Natural mortality ($M$) of wild populations of fish is difficult to measure but is an important input variable into stock assessments. We believe that the estimate of $M$ derived from the maximum-age-based method of *Hewitt & Hoenig (2005)* obtained in this study, $M = 0.32$, was a reasonable estimate for the fully recruited ages in our study, as it compared favorably with the estimate of $M = 0.38$ for black sea bass as used in SEDAR 25 (*SEDAR, 2017b*). Though the estimate of $M$ for black sea bass was derived using the method of Hoenig (1983), when computing $M$ using the *Hewitt & Hoenig (2005)* method, the value was the same. Both species have similar maximum age (11 years for black sea bass) and length. We do not believe this is a suitable estimate of $M$ for all ages, because younger fish are more vulnerable to predation, and thus likely have higher mortality rates. The age-varying $M$ calculated using *Charnov, Gislason & Pope (2013)* seems a more appropriate estimator for the younger ages. The initial *Charnov, Gislason & Pope (2013)* estimates of $M$ starting with the fully recruited age of 3 was approximately equal to the *Hewitt & Hoenig (2005)* estimate, which suggests that the fish's initial fast growth allowed it to reach a size less vulnerable to predation. The age-specific estimates of $M$ for the older ages continue to decrease until stabilizing at 0.17 at age-12 (Table 1). The maximum-age-based method of *Then et al. (2015)* resulted in a point estimate equal to 0.47, while *Then et al.*'s *(2015)* growth-based estimator resulted in a value of 0.03. When deciding which estimator to use when they are so widely varying, the likely reliability of the input to the estimator should be considered. Generally, we do not believe that purely size-based estimates, as in the case of *Then et al.*'s *(2015)* growth-based estimator, are appropriate. *Then et al. (2015)* did caution about using this estimator, which included a high variance about the estimate. Jolthead porgy, compared to other co-occurring reef fish such as species of Lutjanidae, are shorter-lived and occupy a lower trophic level, so we expect $M$ to be higher than estimates for Lutjanids. The estimate of $L_\infty$ is limited by the samples used to develop the model and is tightly, inversely-correlated with $K$. Because of the lack of small fish to accurately describe initial growth at the youngest ages, which affects the trajectory of the growth curve and thus the value of the asymptotic length, we are more confident using the maximum age versus the estimate of $L_\infty$ to estimate $M$. We have some confidence in the maximum age found

in this study because it is similar to what was found for several congeners (Table 2), while our maximum size if 680 mm TL is quite a bit smaller than the largest recorded jolthead porgy (760 mm FL, or 849 mm TL. We recognize the possibility that the maximum age of jolthead is greater than the 13 years that we found, but we have found nothing in the literature to show that is indeed fact.

When estimating survivorship only on the fully recruited ages, this estimate omits a large amount of mortality that occurs on younger fish, thus affecting the survivorship estimate. Few of the fish in our samples were older than 8 yrs (5 of 635). Though our samples are limited for this study, the age frequency suggests that the chance of survivorship to the oldest age may truly be about 1.1%. There is evidence that selectivity of hook and line gear is not dome-shaped for red porgy taken in commercial or headboat fisheries (*SEDAR, 2012*). If true for jolthead porgy, it suggests our study had the potential to collect the largest and oldest fish in the population (but see discussion in the paragraph below). These observations give weight to the argument that the *Then et al. (2015)* age-based method overestimated $M$, and the *Charnov, Gislason & Pope (2013)* estimate of $M$ at age may have under-estimated. Because we believe that the *Hewitt & Hoenig (2005)* estimate for fully recruited ages may be the best estimate, but not appropriate for the smallest, youngest fish, we suggest that the *Charnov, Gislason & Pope (2013)* estimates be scaled to the *Hewitt & Hoenig (2005)* estimate.

Our preliminary estimates of total mortality, $Z = 0.70$, and fishing mortality, $F = 0.38$, are somewhat difficult to interpret. A basic assumption of the catch curve analysis method we used to calculate $Z$ is that the sample is representative of the population. While we feel our sampling fairly represents the fished population in the SEUS, it is possible that the portion of the population vulnerable to the commercial fishery was underrepresented by our sampling. This could result in a biased estimate of maximum age and growth parameters, which would have an effect on our estimates of $M$ and thus $Z$. Managers should thus interpret these results with caution.

This is the first published study of jolthead porgy life history. We documented the first data on reproductive seasonality of the species, showing a discrete spawning season during the late winter months. We have shown that otolith sections of jolthead porgy contain annuli that are remarkably easy to enumerate and that otolith sections are reliable structures for aging. Opaque zones on jolthead porgy sagittae are assumed to be deposited once a year March–June, and the growth trajectory follows a similar path to its smaller congeners. Our estimate of $M$ based on *Hewitt & Hoenig (2005)* is reasonable for a species with moderately fast growth and longevity of at least 13 years. We provide the first preliminary estimates of total and fishing mortality for this species. These results begin to fill an information gap for this data poor species and should allow fishery regulators to set future ACLs that are based on better scientific data such as life history parameters, versus less than optimum data that has been used in the past for data-limited species (e.g., catch histories), as pointed out by *Carruthers et al. (2017)*.

# ACKNOWLEDGEMENTS

We gratefully acknowledge the many NMFS headboat and commercial port samplers over the years whose efforts made this study possible. R Munoz, R Allman, T Kellison and two anonymous reviewers provided reviews which greatly improved the manuscript.

## Funding

This work was supported by National Marine Fisheries Service Marine Fisheries Initiative (MARFIN) grant 15MARFIN INHOUSE008. The external funders, NMFS, have final approval over the decision to publish peer reviewed journal articles, to ensure that the study is good science and well written. The funders had no role in study design, data collection and analysis, decision to publish, or preparation of the manuscript.

## Grant Disclosures

The following grant information was disclosed by the authors:
National Marine Fisheries Service Marine Fisheries Initiative (MARFIN): 15MARFIN INHOUSE008.

## Competing Interests

The authors declare there are no competing interests.

## Author Contributions

- Michael L. Burton conceived and designed the experiments, performed the experiments, analyzed the data, wrote the paper, prepared figures and/or tables, reviewed drafts of the paper.
- Jennifer C. Potts conceived and designed the experiments, performed the experiments, analyzed the data, contributed reagents/materials/analysis tools, wrote the paper, prepared figures and/or tables, reviewed drafts of the paper.
- Jordan Page performed the experiments, sectioning of otolith sections, reading of ages.
- Ariel Poholek conceived and designed the experiments, contributed reagents/materials/analysis tools, collection of majority of samples in the field.

## Animal Ethics

The following information was supplied relating to ethical approvals (i.e., approving body and any reference numbers):

All specimens used in this study were killed as part of legal fishing operations and were already dead when sampled by port agents; thus all research was conducted in accordance with the Animal Welfare Act (AWA) and with the US Government Principles for the Utilization and Care of Vertebrate Animals Used in Testing, Research, and Training (USGP) OSTP CFR, May 20, 1985, Vol. 50, No. 97.

## Data Availability

The raw data has been supplied as a Supplemental File.

## Supplemental Information

Supplemental information for this article can be found online at http://dx.doi.org/10.7717/peerj.3774#supplemental-information.

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
