# Peer review of "Age, growth, mortality and reproductive seasonality of jolthead porgy, Calamus bajonado, from Florida waters"

_PeerJ, doi:10.7717/peerj.3774_

## Round 0.1 · original submission · Minor Revisions

· Academic Editor

Minor Revisions

Both reviewers are supportive of publication pending your consideration of a number of "minor" suggestions. I note that both have raised the issue of further exploration of the age & sex data and I would hope that this is possible.

Reviewer 1 ·

Basic reporting

The article generally well written, and constructed in an appropriate style for scientific publication. Some minor recommendations below.

Lines 72-83 - Reliance on unpublished data to describe the extent of the fishery. I recommend citing at least the final reports from these studies.

Figure 3 – I recommend increasing the size of the otolith images and marking increments with a more contrasting colour, perhaps red.

Line 245 – It would be useful for the reader if the parameter values were presented in sentence form, not just in the equation, i.e., “Linf = 737, k =….”

Lines 368-381 – This paragraph should be disbanded, and individual limitations simply weaved into the discussion point that they relate to. For example, the depth sampling bias arising from the dominance of samples from recreational anglers should be outlined when the ontogenetic depth shift is discussed in the second paragraph of the Discussion.

Experimental design

The specific knowledge gaps, and corresponding goals of the paper, are not well outlined. The authors rely on the generalisation that 'demographic knowledge is useful for fisheries management', but fail to articulate how demographic information will be used in their specific case of jolthead porgy in Florida, particularly with respect to setting the ACL. To address this, I recommend the authors include background information on how particular demographic parameters are used to inform harvest strategies for fished species in the Introduction. This should then allow development of more precise goals for the study, rather than simply, “…to provide the first published information on age-growth parameters, natural mortality rates and reproductive seasonality for jolthead porgy in the SEUS, filling in an important gap for this data-poor species.”

Further clarification of the spatial and temporal extent of sampling is required. The information in lines 212-214 should be provided at the beginning of the Methods, and if samples were collected from multiple locations within each of the two regions, a map should be provided indicating the sample sizes at each location. Monthly sample sizes are already indicated in Figure 1, but a sentence explaining the sampling gap between June and September is required toward the beginning of the Methods (elaborate on lines 207-209; are fish not encountered because it is a seasonal fishery?)

Specific comments:
Line 105-106 – How many of each sex?

Line 126 – Please clarify the plane of sectioning, I assume transverse?

Lines 147-148 – How many individuals were excluded?

Figure 1 – An explanation for the low sample sizes in Graph A relative to Graph B is required.

Lines 162-167 – Calculation of monthly age would also benefit from knowledge of the time between birth and first increment formation. This may result in a reduction in monthly ages.

Lines 211-212 – There is no mention of readability. I don’t think I have encountered a species where every individual was considered sufficiently readable to include in age and growth analyses. Further clarification on quality control procedures with respect to readability (e.g. a readability scoring system) is required in both the Methods and Results sections, as is hinted at in line 232.

Validity of the findings

It’s unclear why demographic parameters were not investigated separately for each sex. If differences exist between the sexes, conclusions made from data pooled across sexes may be seriously compromised. Potential differences between males and females must be explored before the values obtained in the current study can be used for fisheries management, unless the authors can provide a suitable explanation for this apparent oversight. Minor comments below.

The Discussion generally suffers from restatement of results and few conclusions regarding fisheries implications. The authors need to indicate what each result means for the vulnerability (or not) of the population to fishing, and how this might be appropriately managed.

Lines 305-306 – Some comment on the likely depth range of samples in the current study is therefore required here, based on known fishing areas for commercial and recreational fishers in Florida. This will indicate whether the current experimental design was sufficient to sample the largest(oldest) individuals in the population, and has implications for natural mortality and longevity estimates.

Lines 319-323 – I recommend rewriting this sentence. To the reader, the authors appear to suggest that the Hewitt& Hoenig (2005) mortality estimate is supported by comparison with a different species, whose mortality was estimated using a different method (Hoenig 1983). This is clearly illogical, but I expect it can be clarified using different wording.

Lines 353-357 – This should be moved to the second paragraph of the Discussion where the theory on ontogenetic depth shifts is proposed.

Line 385 – I recommend changing the phrase “relatively easy” to “one of the easiest”, given inter-reader comparisons and the fact that no individuals were excluded on the basis of poor readability.

Line 389 - This is a bit of a ‘throwaway’ statement, which reaffirms my concern regarding the lack of specific goals in this study. The authors need to specify how their results might be used to refine the management of jolthead porgy, otherwise the relevance of the study is not clear to the reader.

Additional comments

I'm surprised the authors did not attempt to estimate fishing mortality, given the relevance of such an estimate to fisheries management and the recent breach of the ACL for jolthead porgy (potentially indicating intense fishing pressure). An estimate of fishing mortality could be calculated from existing data by subtracting the value of natural mortality from total mortality, the latter of which was not estimated by the authors, but could be easily obtained from the slope of the descending limb of the age-frequency curve.

·

Basic reporting

The paper meets the PeerJ standards of basic reporting. See related comments below.

Experimental design

The paper meets the PeerJ standards of experimental design. See related comments below.

Validity of the findings

The paper meets the PeerJ standards of validity of findings. See related comments below.

Additional comments

General comments

The authors aged a moderate-sized sample of jolthead porgy collected from fishery-dependent sources in Florida. They then used these ages to estimate growth and mortality rate parameters. Additionally, the authors provided information on the temporal pattern in spawning for this species based on macroscopic gonad information. All of this information is novel for jolthead porgy in this region. The paper meets all of the PeerJ criteria for publication and we recommend acceptance after addressing the points raised below.

Line 74-83 - The FL headboats land 62% of jolthead porgy on the US east coast which means (we assume) that a moderate amount is landed in states to the north. Yet, the commercial landings are 100% Florida and MRIP shows 98% landed in FL waters. Are the MRIP and commercial numbers correct or do they result from a species identification problem or lumping (e.g. porgies complex)? Please comment on whether or not the fishery-dependent data match with fishery-independent (e.g. MARMAP) information on their distribution. It would be nice to have a little more evidence that the focus on collections in Florida is a sound one given the headboat vs mrip/commercial discrepancy.

The authors mention that 30% of samples came from Ft Pierce – Miami, and remaining 70% came from FL Keys. Were there differences between these two populations (e.g. sex ratio, growth parameters). Do the authors suspect there would exist differences between these southern samples and samples to the north?

Line 178-179/line 184-185 – since the fish for aging were collected well after the fishery had begun, the tmax could be an underestimate for this species. This is brought up in Discussion but it would be nice to have further outside information on jolthead maximum sizes (e.g. world record? Other?) or max ages even if from a different region. It would help you and the reader gauge whether your tmax is biased low or not. Based on K and Linf, joltheads look to have a different life history strategy relative to the other Calamus spp. and the higher Linf appears to be justified given larger max sizes. Given this, one might expect the tmax to be different for this species. Using tmax approaches to estimate M results in M estimates for jolthead that are similar to the other Calamus spp. which doesn’t seem quite right given the larger sizes at age (i.e. less predation mortality because larger) for joltheads relative to the other Calamus spp. Some discussion on this or perhaps estimating M with a slightly higher (but justified) tmax could be considered.

Line 194-199 – These estimates of survivorship would be without fishing, correct? Please clarify in Methods and in Discussion. This is related to assumptions of what tmax represents (recent tmax after fishing vs tmax of unfished population). Please present the results for the survivorship analysis in Results section.

Line 353-359 – These are lines from Discussion that relate to our comments about tmax and M. The fact that fishing has likely truncated the age distribution seems an equal (or more important) bias to tmax than the spatial location of your samples. This 1% survivorship estimated with M from equation does not take into account F that is also resulting in the lower tmax.

Line 71 brings up hermaphroditism. No mention later of length/age of transition. Given that sex was determined, it would be nice to see data on this.

Copy edits/specific suggestions

Line 76: number of fish presented as 1886 (no comma) while in Line 80, number of fish presented as 73,381 (with comma)
Line 123: not necessary to specify
Line 223: delete word “snapper”
Line 289: confirm species attribution “Linnaeus 1978.” Fishbase has Linnaeus 1758.
Line 305: suggest editing “all four species of Calamus” to “all four species discussed here” or similar, as there are more than four total species in this genus. Line 312 reflects this already.
Line 363: “over-estimated” to “overestimated”
Line 207-209 – If that is the case for samples in summer, then where did June to September otoliths come from for edge analysis described two paragraphs down and presented on Figure 3?
Table 1 – Please provide estimate of variance around predicted TL.
Table 1 legend – Please provide a brief description of where predicted TL and M’s came from (VBGF, Hewitt and Hoenig)
Figure 1 legend: Provide sampling location and years as in Figure 2 legend…same comment for Figure 4.
Figure 2 caption – For the % of previous increment text, please clarify that this is translucent zone (I assume?). Throughout manuscript there is use of both “increment” and “zone” when referring to opaque or translucent regions of otoliths. If “increment” and “zone” have different definitions then please define in Methods; if not, then please use only one of these terms for consistency throughout.
Figure 4: legend symbol for Observed Size at Age is potentially confusing as it may be interpreted as part of the plot at first glance. Perhaps shifting the legend to the right or placing it inside a box would be less ambiguous.

Nice work,

Jeff Buckel & Brendan Runde

NC State University

---

## Round 0.2 · accepted · Accept

· Academic Editor

Accept

Thank you for a thoughtful and timely revision of the manuscript.